# Challenges for Precise Subtyping and Sequencing of a H5N1 Clade 2.3.4.4b Highly Pathogenic Avian Influenza Virus Isolated in Japan in the 2022–2023 Season Using Classical Serological and Molecular Methods

**DOI:** 10.3390/v15112274

**Published:** 2023-11-18

**Authors:** James G. Komu, Hiep Dinh Nguyen, Yohei Takeda, Shinya Fukumoto, Kunitoshi Imai, Hitoshi Takemae, Tetsuya Mizutani, Haruko Ogawa

**Affiliations:** 1Graduate School of Animal and Veterinary Sciences and Agriculture, Obihiro University of Agriculture and Veterinary Medicine, 2-11 Inada, Obihiro 080-8555, Hokkaido, Japan; jkomu@jkuat.ac.ke (J.G.K.); ndhiep2412@gmail.com (H.D.N.); 2Department of Medical Laboratory Sciences, College of Health Sciences, Jomo Kenyatta University of Agriculture and Technology, Nairobi P.O. Box 62000-00200, Kenya; 3Department of Veterinary Medicine, Obihiro University of Agriculture and Veterinary Medicine, 2-11 Inada, Obihiro 080-8555, Hokkaido, Japan; ytakeda@obihiro.ac.jp (Y.T.); fukumoto@obihiro.ac.jp (S.F.); imaiku@obihiro.ac.jp (K.I.); 4Research Center for Global Agromedicine, Obihiro University of Agriculture and Veterinary Medicine, 2-11 Inada, Obihiro 080-8555, Hokkaido, Japan; 5National Research Center for Protozoan Diseases, Obihiro University of Agriculture and Veterinary Medicine, 2-11 Inada, Obihiro 080-8555, Hokkaido, Japan; 6Center for Infectious Diseases Epidemiology and Prevention Research, CEPiR, Tokyo University of Agriculture and Technology, Fuchu-shi 183-8509, Tokyo, Japan; fy7210@go.tuat.ac.jp (H.T.); tmizutan@cc.tuat.ac.jp (T.M.)

**Keywords:** highly pathogenic avian influenza, H5N1, classical diagnostic methods, antigenic and genetic evolution

## Abstract

The continuous evolution of H5Nx highly pathogenic avian influenza viruses (HPAIVs) is a major concern for accurate diagnosis. We encountered some challenges in subtyping and sequencing a recently isolated H5N1 HPAIV strain using classical diagnostic methods. Oropharyngeal, conjunctival, and cloacal swabs collected from a dead white-tailed eagle (*Haliaeetus albicilla albicilla)* were screened via real-time RT-PCR targeting the influenza A virus matrix (M) gene, followed by virus isolation. The hemagglutination inhibition test was applied in order to subtype and antigenically characterize the isolate using anti-A/duck/Hong Kong/820/80 (H5N3) reference serum or anti-H5N1 cross-clade monoclonal antibodies (mAbs). Sequencing using previously reported universal primers was attempted in order to analyze the full-length hemagglutinin (HA) gene. Oropharyngeal and conjunctival samples were positive for the M gene, and high hemagglutination titers were detected in inoculated eggs. However, its hemagglutination activity was not inhibited by the reference serum or mAbs. The antiserum to a recently isolated H5N1 clade 2.3.4.4b strain inhibited our isolate but not older strains. A homologous sequence in the previously reported forward primer and HA2 region in our isolate led to partial HA gene amplification. Finally, next-generation sequencing confirmed the isolate as H5N1 clade 2.3.4.4b HPAIV, with genetic similarity to H5N1 strains circulating in Japan since November 2021.

## 1. Introduction

The H5 subtype of highly pathogenic avian influenza viruses (HPAIVs) is a global menace because of its high morbidity and mortality rates in poultry and potential for zoonotic transmission to humans [1]. Although the earliest ancestral strain of H5 subtype HPAIVs, namely A/goose/Guangdong/1/1996 (H5N1) (Gs/GD), was originally described in China in 1996, the steady circulation of these strains in poultry began after the concurrent 2003–2004 winter season outbreak of the Asian-lineage H5N1 virus in eight East and Southeast Asian countries, including Japan [2,3]. The hemagglutinin (HA) genes of these viruses have consistently and rapidly evolved, leading to their classification into 10 first-order clades (clades 0–9); some clades have further diversified into fifth-order subclades [4]. Notably, the neuraminidase (NA) gene of H5N1 clade 2.3.4.4 HPAIVs has been genetically reassorting with other avian influenza virus (AIV) NA genes, generating different H5Nx HPAIV subtypes since 2008. H5Nx HPAIVs have spread to Asia, Europe, North America, and Africa, undoubtedly through wild migratory birds [5,6].

Although the number of documented cases of H5N1 outbreaks has significantly decreased since the early 2010s, their resurgence through wild-bird-adapted H5N1 strains and subsequent predominance globally has been evidenced since late 2021 [7]. In Japan, the first H5N1 resurgence was reported in a layer chicken farm in November 2021 in the southern part of the country, where H5N8 was co-circulating. The H5N1 strain carried a similar HA gene as H5N8 circulating in Europe during the same period [8]. Coincidentally, a slightly divergent phylogenetic cluster with similar genes to those of H5N1 circulating in Europe and North America was also reported in wild birds, an Ezo red fox and a Japanese raccoon dog in Northern Japan [9,10]. This indicated the possibility of the simultaneous introduction and circulation of different genotypes of H5N1 in Japan. Subsequently, poultry outbreaks have occurred in multiple Japanese prefectures, leading to the culling of a record 17.7 million birds in the 2022–2023 season [11].

Recently, next-generation sequencing (NGS) has been applied for the simultaneous subtyping and sequencing of AIVs. However, as high costs preclude the use of NGS for routine surveillance, classical serological subtyping methods, including hemagglutination inhibition (HI) and NA inhibition assays, remain reliable, affordable, and accepted serological tools for global influenza surveillance, as recommended by the World Health Organization (WHO Manual) [12]. For HA subtyping, the HI test utilizes subtype-specific reference sera to react with specific HA subtypes, thereby inhibiting their hemagglutination activities. In addition, several HA and NA molecular subtyping methods using various primer sets are available [13,14,15,16,17]. In particular, universal primer sets that anneal the non-coding regions of the eight segments of influenza viruses’ genes have been widely applied for full-length amplification and subsequent Sanger’s sequencing, as described by Hoffmann et al. [18]. The sequences recognized using Hoffmann’s primers are highly conserved among all segments with only 2–3 segment-specific nucleotides at the ends. With the continuous evolution of H5Nx viruses, especially changes in the HA gene, a series of antigenically and genetically diverse isolates is anticipated. It is therefore paramount to continuously confirm the effectiveness of classical serological and molecular diagnostic methods for the accurate detection of emerging AIV strains and update them consistently.

According to the Japanese Ministry of the Environment (MoE), 242 cases of H5-subtype HPAIV infections were detected among wild birds between 25 September 2022 and 19 April 2023 [19]. Among them, one case involved a dead white-tailed eagle found by our group in Urahoro, Hokkaido prefecture, on 23 November 2022; the animal was reported to the MoE and confirmed positive for H5N1 HPAIV in the authorized MoE diagnostic laboratory. Concomitantly, oropharyngeal, conjunctival, and cloacal swabs were collected for further antigenic and genetic analyses in our laboratory. In this study, we report the challenges experienced in precisely subtyping and sequencing the isolate using classical diagnostic methods and describe possible solutions to these shortcomings.

## 2. Materials and Methods

### 2.1. Sample Collection and Virus Isolation

Oropharyngeal, conjunctival, and cloacal swabs were separately collected in viral transport medium from a dead white-tailed eagle (*Haliaeetus albicilla albicilla)* found in Urahoro, Hokkaido, Japan. The viral transport medium consisted of minimum essential medium (Nissui Pharmaceutical Co., Ltd., Tokyo, Japan) supplemented with 0.5% bovine serum albumin (FUJIFILM Wako Pure Chemical Co., Osaka, Japan), 0.15% NaHCO_3_ (FUJIFILM Wako Pure Chemical Co.), 1 mg/mL of kanamycin (Meiji Seika Pharma Co., Ltd., Tokyo, Japan), 100 µg/mL of gentamycin (MSD Japan, Tokyo, Japan), and 5 µg/mL of amphotericin B (Bristol-Myers Squibb Co., New York, NY, USA) according to the WHO Manual [12], with some modifications. The bird was identified as a juvenile white-tailed eagle (gender unknown) at the Institute for Raptor Biomedicine Japan (Kushiro, Japan). The swab samples were inoculated into the allantoic cavities of embryonated chicken eggs for virus isolation, followed by the hemagglutination test, as described by the WHO Manual [12].

### 2.2. RNA Extraction, Real-Time RT-PCR, and Subtype-Specific RT-PCR

Viral RNA was extracted from the swab samples or allantoic fluid harvested from the inoculated embryonated chicken eggs using a QIAamp Viral RNA Kit (Qiagen, Hilden, Germany), according to the manufacturer’s instructions. cDNA was synthesized using random hexamer primers (Invitrogen, Carlsbad, CA, USA) and M-MLV Reverse Transcriptase (Invitrogen) under the following conditions: 25 °C for 10 min, 37 °C for 60 min, and 65 °C for 10 min. Real-time PCR targeting the influenza A virus matrix (M) gene [20] was performed as previously described. HA molecular subtyping was performed using 15 subtype-specific primer sets (H1–H15), as described previously [17].

### 2.3. HA Subtyping and Antigenic Characterization by the HI Test

Fifteen HA subtype-specific antisera kindly provided by the OIE Reference Laboratory for AIV at Hokkaido University (Sapporo, Japan) were used for HA subtyping (Appendix A). For antigenic characterization using the HI test, the reactivity of the isolate to anti-H5 sera, including the reference serum (anti-A/duck/Hong Kong/820/80, H5N3) and anti-A/chicken/Miyazaki/K11/07 (H5N1) serum, kindly provided by the National Institute of Animal Health (Tsukuba, Japan), was compared to that of other H5-subtype viruses available in our laboratory. Similarly, we used three neutralizing anti-HA monoclonal antibodies (mAbs) previously produced in our laboratory. These mAbs included two anti-H5 cross-clade mAbs (3B5.1 and 3B5.2) produced from the A/chicken/Yamaguchi/7/04 (H5N1) of clade 2.5, while 1G5 was produced from the A/chicken/Miyazaki/K11/07 (H5N1) of clade 2.2 and was reactive against strains of clades 1, 2.2, 2.3.4 and 2.5 [21]. In addition, we used anti-H5 serum newly prepared by the OIE Reference Laboratory against the recent H5N1 HPAIV clade 2.3.4.4b isolate A/white-tailed eagle/Hokkaido/22-RU-WTE-2/2022 [10]. The HI test was performed according to the WHO Manual [12].

### 2.4. Amplification and Sanger Sequencing of the HA Gene Using Hoffmann’s Primers

cDNA was synthesized from the RNA samples extracted from the allantoic fluid using Uni-12 primers and Superscript III Reverse Transcriptase (Invitrogen). The cDNA was used as a template to amplify the full-length HA gene using the primers (forward [Bm-HA-1], 5′-TATTCGTCTCAGGGAGCAAAAGCAGGGG-3′; reverse [Bm-NS-890R], 5′-ATATCGTCTCGTATTAGTAGAAACAAGGGTGTTTT-3′) and protocol described by Hoffmann et al. [18]. We also designed and amplified the region using a modified version of Hoffmann’s HA forward primer (5′-TATTCGTCTCAGGGAGCAAAAGCAGGGGT-3′). The obtained PCR products were electrophoresed on a 1.5% agarose gel stained with Midori Green Xtra (Nippon Genetics, Tokyo, Japan) and visualized using FastGene FAS-Digi PRO (Nippon Genetics). The amplicon was excised from the gel, purified using a QIAquick PCR Purification Kit (Qiagen), and sequenced using a BigDye terminator ver. 3.1 Cycle Sequencing Kit (Thermo Fisher Scientific Inc., Waltham, MA, USA), according to the manufacturer’s instructions. The sequences were read using an Applied Biosystems 3500 Genetic Analyzer (Thermo Fisher Scientific).

### 2.5. NGS and Phylogenetic Analysis

The NEBNext Ultra II RNA Library Prep Kit for Illumina (New England Biolabs, Ipswich, MA, USA) was used to construct cDNA libraries for deep sequencing, according to the manufacturer’s instructions. The library quality was assessed on a Qubit^®^ 4.0 Fluorimeter (Invitrogen). A MiniSeq benchtop sequencer (Illumina, San Diego, CA, USA) was used for deep sequencing with the paired-end reads of 151 nucleotides. FASTQ-formatted sequences were generated using MiniSeq Local Run Manager (Illumina) and imported into CLC Genomics Workbench 12.0.3 (CLC bio, Aarhus, Denmark). Thereafter, the sequence data were trimmed, and low-quality sequences were removed. Subsequently, the processed sequence data were assembled into contigs using the de novo assembly command in CLC Genomics Workbench (CLC bio). The trimmed sequences were mapped to known representative HPAIV genome sequences using the Map Read to Reference command of CLC software (CLC Genomics Workbench 12.0.3), and consensus sequences were obtained. To construct phylogenetic trees for all eight influenza A virus gene segments, related sequences were downloaded from GenBank and aligned, and their evolutionary history was inferred using the maximum likelihood method and Tamura–Nei model [22] in Mega software version 11.0.13 [23]. The validity of the phylogenetic trees was evaluated using 1000 bootstrap replicates.

## 3. Results

### 3.1. Failed HA Subtyping and Full-Length HA Gene Amplification of the H5N1 Isolate by Classical Methods

Real-time RT-PCR of the influenza A virus M gene gave borderline positive cycle threshold (Ct) values ranging from 33.51 to 36.35. Upon propagation in embryonated chicken eggs, the allantoic fluid harvested from the eggs inoculated with the oropharyngeal and conjunctival swabs had HA titers of 256 and 512, respectively (Table 1). Notably, the allantoic fluid obtained from the cloacal sample-inoculated eggs did not exhibit HA activity. Therefore, it was not used for subsequent serological and molecular analyses. Real-time RT-PCR using RNA extracted from the allantoic fluid of the eggs inoculated with oropharyngeal and conjunctival swabs gave Ct values of 21.81 and 20.93, respectively, suggesting the isolation of influenza A virus (Table 1). Unexpectedly, classical subtyping of the HA-positive allantoic fluid via the HI test using 15 HA subtype-specific reference antisera (Appendix A) produced negative results. Conversely, HA molecular subtyping of both the oropharyngeal and conjunctival samples via conventional RT-PCR [17] identified the isolate as an H5-subtype strain (Figure 1A).

Furthermore, RT-PCR of the full-length HA gene using Hoffmann’s primers resulted in the amplification of an approximately 658-bp band, which was much shorter than the expected full-length HA amplicon (approximately 1807 bp) for both samples (Figure 1B). Sanger sequencing of the amplicon using the forward primer and subsequent BLAST analysis (https://blast.ncbi.nlm.nih.gov/Blast.cgi, accessed on 3 February 2023) revealed that the amplicon was a partial HA gene aligning to the HA2 region. Because of difficulties in subtyping the isolate using the HI test and full-length amplification of the HA gene, the isolate was submitted to NGS.

### 3.2. Genetic Characterization and Phylogenetic Analysis

The entire genomic sequence of the H5N1 isolate was compared to that of other H5Nx isolates available in GenBank or the Global Initiative on Sharing All Influenza Data (GISAID), and our isolate was confirmed to be HPAIV of the H5N1 subtype. The isolate was designated as A/white-tailed eagle/Japan/OU-1/2022 (WTE/Jp) (H5N1). BLAST analysis revealed that all eight gene segments were closely related to those of other H5N1 viruses circulating in Japan since November 2021 (nucleotide identity ≥ 99.39%; Appendix A). Consistent with other HPAIVs, WTE/Jp (H5N1) had a PLRERRRKR/GLF amino acid motif at the HA cleavage site. Phylogenetic analysis of the HA gene confirmed that WTE/Jp (H5N1) belonged to clade 2.3.4.4b and clustered with other H5N1 HPAIV strains that had been circulating in domestic poultry (chickens and emus) and wild birds in Hokkaido and other prefectures in Japan (Figure 2). Similar phylogenetic clustering was observed for the NA gene and six other internal genes (Appendix A). The nucleotide sequences of the eight gene segments were deposited in GenBank under the accession numbers LC775576–LC775583.

### 3.3. Reason for the Failed Full-Length Amplification of the HA Gene

To clarify why the amplification of the full-length HA gene was unsuccessful using Hoffmann’s primers, the HA gene sequence of WTE/Jp (H5N1) obtained via NGS was further analyzed. The eight-nucleotide sequence “AGCAGGGG” at positions 1121–1128 in the HA coding region of WTE/Jp (H5N1) was identical to the last eight nucleotides at the 3′-end of Hoffmann’s HA forward primer (Figure 3). This most likely led to preferential annealing to the HA2 region and the amplification of the 658-bp partial HA gene via RT-PCR. To confirm whether this phenomenon was unique to WTE/Jp (H5N1), the sequence data were aligned with those of other recent H5N1 HPAIV clade 2.3.4.4b isolates (group i), H5Nx isolates belonging to different subclades that had circulated during or before the 2021–2022 season (group ii), and nonGs/GD-lineage H5 low-pathogenic AIVs (group iii), in descending order of the year of occurrence. The observed eight-nucleotide sequence at the HA2 region, resulting in the failure of amplification of the full-length HA gene of WTE/Jp (H5N1), was shared among all aligned H5N1 HPAIV isolate sequences in 2021–2023, but this was infrequently found among the older isolates (Figure 3). To clarify why the amplification of the full-length HA gene failed, we modified Hoffmann’s HA forward primer (5′-TATTCGTCTCAGGGAGCAAAAGCAGGGG-3′) by adding one T residue at the 3′-end. Using the modified HA forward primer (5′-TATTCGTCTCAGGGAGCAAAAGCAGGGGT-3′) in PCR, the full-length HA gene of AIV isolates with prominent partial HA gene amplification using Hoffmann’s original primer, including WTE/Jp (H5N1) and other strains (Figure 4A), was successfully amplified (Figure 4B).

### 3.4. Antigenic Characterization

The antigenic profile of WTE/Jp (H5N1) was compared to that of other H5 subtype viruses belonging to different clades via the HI test using multiple anti-H5 subtype sera and cross-clade mAbs. The hemagglutination activity of WTE/Jp (H5N1) was not inhibited by anti-A/duck/Hong Kong/820/80 (H5N3) reference serum (HI titer < 10), but it was weakly inhibited by anti-A/chicken/Miyazaki/K11/07 (H5N1) serum (HI titer = 80). Similarly, anti-H5 subtype cross-clade mAbs had no inhibitory activity (Table 2). Nevertheless, newly produced antiserum against A/white-tailed eagle/Hokkaido/22-RU-WTE-2/2022 clade 2.3.4.4b isolated in the 2021–2022 season strongly inhibited the hemagglutination activity of WTE/Jp (H5N1) at an HI titer of 1280. This antiserum against A/white-tailed eagle/Hokkaido/22-RU-WTE-2/2022 moderately inhibited the EA-nonGs/GD strain (A/duck/Hong Kong/820/80, H5N3) (HI titer = 320) and weakly inhibited the H5N1 HPAIVs isolates obtained from chickens in 2004–2007 (HI titer = 80); however, it failed to inhibit the clade 2.3.2.1 HPAIV H5N1 strain isolated from a whooper swan in 2011 (Table 2). These results suggest a remarkable antigenic evolution in the recent H5N1 isolates, including WTE/Jp (H5N1), compared to older strains previously reported in Japan.

## 4. Discussion

Japan, similar to many other countries, has invariably been affected by H5-subtype HPAIV outbreaks, causing devastating economic losses in the poultry industry. The H5N1 HPAIV outbreak in the 2022–2023 season unanticipatedly led to the culling of a record number of birds [11]. Similarly, a sizable number of wild bird cases were reported [19], some of which resulted in anomalous mortalities, especially in crows, as observed in the 2021–2022 season [10]. These cases included the dead white-tailed eagle described in this paper, from which we isolated an H5N1 HPAIV strain from oropharyngeal and conjunctival samples. According to the Japan Red List 2020, white-tailed eagles have been classified as a vulnerable bird species [24] and several strategies have been put in place to ensure their conservation. Due to strict legal and biosafety control, we were not allowed to collect further samples for postmortem analysis and investigate the viral titers in different organs. However, white-tailed eagles are reportedly relatively susceptible to H5Nx clade 2.3.4.4b HPAIV infections, with cases resulting in high mortality [25,26,27]. In a recent related study conducted in Japan, both natural and experimental infections of white-tailed eagles with clade 2.3.4.4 H5Nx HPAIVs demonstrated the susceptibility of the birds to these viruses resulting in systemic infections. Furthermore, the indirect spread of the infection between white-tailed eagles has also been documented [26].

In this study, despite successful white-tailed-eagle-isolated HPAIV sequencing and subtyping using NGS, our preliminary attempts to serologically subtype and characterize the isolate using anti-A/duck/Hong Kong/820/80, anti-A/chicken/Miyazaki/K11/07, and anti-H5 subtype cross-clade mAbs [21] via the HI test [12] were ineffectual. Similarly, the amplification of the full-length HA gene using widely used universal primers [18] was unsuccessful. We thus primarily focused on reporting these diagnostic challenges and providing possible strategies to counter them. NGS might permit the scalable, rapid subtyping and sequencing of emerging HPAIVs, sequentially facilitating key decisions regarding control measures, vaccine design, and antiviral use [28]. Notwithstanding, NGS is not always affordable for the routine surveillance and characterization of field samples and experimentation in many laboratories. Therefore, the utility of classical diagnostic methods is analogously important, and they should continuously be updated for the precise diagnosis of emerging strains.

Conventionally, cloacal (or fecal) and tracheal samples have been widely used for AIV surveillance and isolation because they predominantly replicate and shed in the gastrointestinal and respiratory systems. However, because HPAIV HA protein can be cleaved by ubiquitous proteases of the furin enzyme family, in addition to trypsin-like enzymes similar to those that cleave lowly pathogenic AIV HA, they easily disseminate, infect, and replicate in other organs [29]. In this regard, the choice of the most appropriate sample type for HPAIV diagnosis is paramount. Recently, compelling evidence, including that in our current study and prior research, has suggested that HPAIVs are likely to be shed in high amounts in the conjunctiva [30,31,32,33], making them reliable for the detection of these viruses, juxtaposed to cloacal swabs.

Phylogenetic analysis revealed that the current H5N1 isolate belongs to clade 2.3.4.4b, and that it was genetically similar to other H5 strains circulating in Japan in the 2022–2023 season and those reported in the 2021–2022 season [10], suggesting the continuous invasion and transmission of H5N1 HPAIVs in Japan. A representative strain in this phylogenetic cluster (A/crow/Hokkaido/0103B065/2022 (H5N1); highlighted in green in the phylogenetic tree) is reportedly highly pathogenic to chickens in vivo [10]. In this study, despite the real-time RT-PCR and HA test data cogently suggesting the presence of influenza A virus in the sample, we could not serologically subtype the isolate using antisera against the reference H1–H15 subtype viruses that we have been using to subtype influenza A viruses. Therefore, after confirming the isolate as H5N1 HPAIV via NGS, we compared its reactivity with that of other H5-subtype viruses against anti-H5 antibodies in different clades, including anti-H5 sera and mAbs, in our laboratory. The hemagglutination activity of WTE/Jp (H5N1) was not inhibited by the reference antiserum against A/duck/Hong Kong/820/80 (H5N3), but it was weakly inhibited by that against A/chicken/Miyazaki/K11/07 (H5N1), even though the antisera were effective against the compared H5-subtype strains. However, it is notable that the A/whooper swan/Hamanaka/2011 (H5N1) subclade 2.3.2.1 was weakly inhibited by the two previously produced antisera used in this study. Correspondingly, newly produced antiserum against the A/white-tailed eagle/Hokkaido/22-RU-WTE-2/2022 strain isolated in 2022 provided by the OIE Reference Lab failed to inhibit the A/whooper swan/Hamanaka/2011 (H5N1) and only weakly or moderately inhibited the other previous strains, despite strongly inhibiting the WTE/Jp (H5N1) strain. Thus, the results of the HI tests clearly indicated that the reactivities of antisera induced by older H5-subtype strains were weak or non-detectable against the H5N1 strain in 2022, whereas the reactivity of the new anti-H5 serum induced by the H5 strain isolated in 2022 against older heterologous H5 strains varied from moderate to non-detectable. Because antisera contain polyclonal antibodies recognizing several antigenic epitopes, non-reactivity suggested a remarkable antigenic evolution in the WTE/Jp (H5N1) isolate. The mAbs used in this study were produced against A/chicken/Miyazaki/K11/07 (H5N1) of clade 2.2 and A/chicken/Yamaguchi/7/04 (H5N1) of clade 2.5, and they were likely to recognize a highly conserved epitope at amino acid residues 158–170 (PTIKRSYNNTNQE), based on the studies on escape mutants [21]. However, WTE/Jp (H5N1) had a substitution (R162I) within its epitope that could have hindered the HI activity of the mAbs. Identically, the mAbs produced against A/chicken/Kumamoto/1-7/2014 (H5N8) clade 2.3.4.4 HPAIV exhibited different reactivity patterns with the viruses of older clades, indicating the divergent antigenicity of clade 2.3.4.4 viruses [34].

RT-PCR using Hoffmann’s primers failed to amplify the full-length HA gene, and instead, a partial HA gene sequence was amplified, contrary to our expectations. Sequence analysis revealed that this was caused by an eight-nucleotide sequence in the HA2 region of our isolate that resembled the last eight nucleotides in the 3′-end of Hoffmann’s forward primer sequence. Mutations leading to nucleotide mismatches between previously designed AIV primers/probe sequences and the HA and/or M genes of emerging variants were previously reported to affect the sensitivity of real-time RT-PCR [35,36,37]. However, to the best of our knowledge, no previous reports have described the failed amplification of the full-length HA gene (or other genes) using Hoffmann’s primers. Remarkably, in addition to the occurrence of this phenomenon in H5N1 clade 2.3.4.4b HPAIVs, it was irregularly observed among some older isolates, raising concerns about the current and past sensitivity of these primers. However, we solved this problem by adding one nucleotide (T) at the 3′-end of the HA forward primer, according to the finding that T is conserved at this position among all isolates aligned to date, which increased the annealing of the primer to the expected region.

The limitation of this study is that the results are based on a single H5N1 HPAIV clade 2.3.4.4b isolate. Nevertheless, we distinctly demonstrated that the current isolate would only be inhibited by newly produced antisera against the A/white-tailed eagle/Hokkaido/22-RU-WTE-2/2022. Moreover, having compared the sequence data of the current isolate with those of other isolates recently circulating in the 2022–2023 season, we think that the failure of Hoffmann’s primers to amplify the full-length HA gene of this isolate is not unique to our laboratory. Sharing this information would potentially help to better classically diagnose HPAIVs and subsequently control their circulation in nature.

## 5. Conclusions

This study demonstrated that continuous antigenic and genetic evolution among emerging H5Nx HPAIVs interferes with the sensitivity of practically available classical diagnostic methods. Of particular note was the failure of the anti-H5 reference serum and mAbs produced against older H5 subtypes to inhibit the currently circulating H5N1 clade 2.3.4.4b HPAIV isolated in this study. However, the antiserum against the recent strain weakly or moderately inhibited the older H5 subtype strains. Additionally, the performance of the universal primers widely used for influenza viruses in the full-length HA gene amplification was compromised by the similarity of the forward primers sequence to a sequence in the HA2 region. Therefore, we suggest the continuous monitoring and updating of classical AIV diagnostic methods to overcome concerns regarding the inaccurate diagnoses of emerging HPAIV strains.

## Figures and Tables

**Figure 1 viruses-15-02274-f001:**
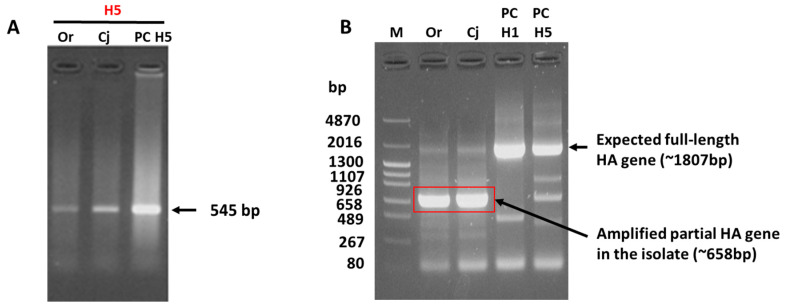
Molecular detection of AIV genes via RT-PCR. PCR products were subjected to agarose gel electrophoresis. (**A**) Subtype-specific RT-PCR for HA subtyping revealing specific amplification of the H5 gene (545 bp) for the oropharyngeal (Or) and conjunctival (Cj) swabs of WTE/Jp (H5N1), similar to the positive control (PC). All the other RT-PCR tests for subtypes H1–H4 and H6–H15 resulted in no amplification. (**B**) Amplification of the HA gene via RT-PCR using Hoffmann’s primers. PCR of the WTE/Jp (H5N1) samples resulted in the amplification of an approximately 658-bp product (highlighted by the red box), contrary to the expected size of approximately 1807 bp for the full-length HA gene, which was confirmed using the positive controls. M: DNA size marker; PC H1: A/Puerto Rico/8/1934 (H1N1); PC H5: A/duck/Hong Kong 820/80 (H5N3).

**Figure 2 viruses-15-02274-f002:**
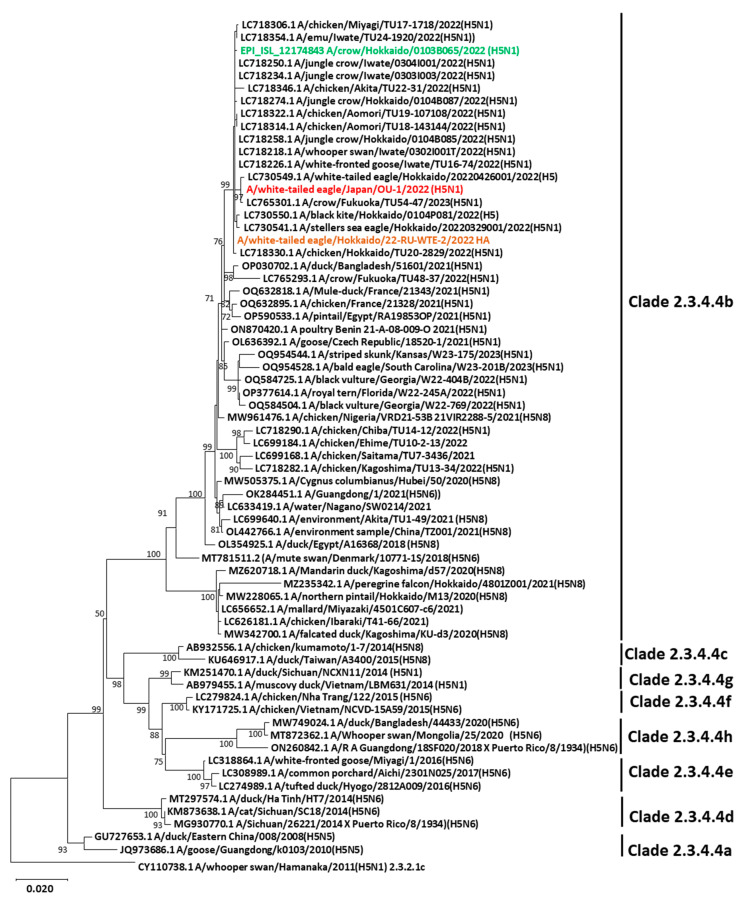
Phylogenetic tree of the HA gene of H5Nx viruses in clade 2.3.4.4. The nucleotide sequence of the HA gene of WTE/Jp (H5N1) and other sequences belonging to different 2.3.4.4 subclades obtained from GenBank and GISAID were used in the phylogenetic analysis. The tree was rooted by A/whooper swan/Hamanaka/2011 (H5N1) belonging to clade 2.3.2.1. Strains in red, brown, and green indicate the H5N1 HPAIV strain isolated in this study, the isolate used to produce an anti-H5 serum that successfully inhibited our isolate, and a representative strain of this phylogenetic cluster previously used to investigate pathogenicity against chickens, respectively. The evolutionary history was inferred using the maximum likelihood method and Tamura–Nei model. The tree was drawn to scale, with branch lengths measured in the number of substitutions per site. Evolutionary analyses were conducted using MEGA11 software version 11.0.13.

**Figure 3 viruses-15-02274-f003:**
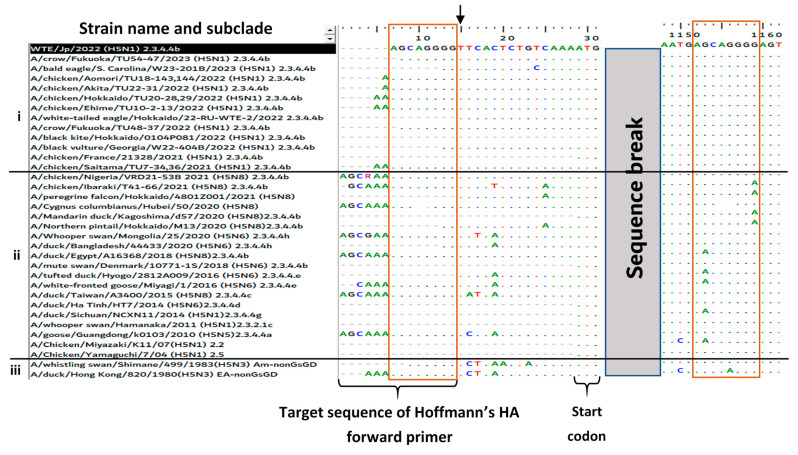
Multiple nucleotide alignment of the HA gene of the WTE/Jp (H5N1) isolate (highlighted in black) with (i) other H5N1 HPAIV strains in clade 2.3.4.4b detected in 2021–2023, (ii) H5Nx strains detected in/before November 2021, and (iii) nonGs/GD-lineage H5 low-pathogenic AIVs, in descending order of the year of occurrence. The gene region covered by Hoffmann’s HA forward primer is indicated under the alignment. The highlighted nucleotides in the two brown boxes, including one sequence in the non-coding region covering the last eight nucleotides in the HA forward primer and the other in the HA2 region, had similar sequences. The black arrow indicates a conserved nucleotide “T” that was added to the modified forward primer described in this paper.

**Figure 4 viruses-15-02274-f004:**
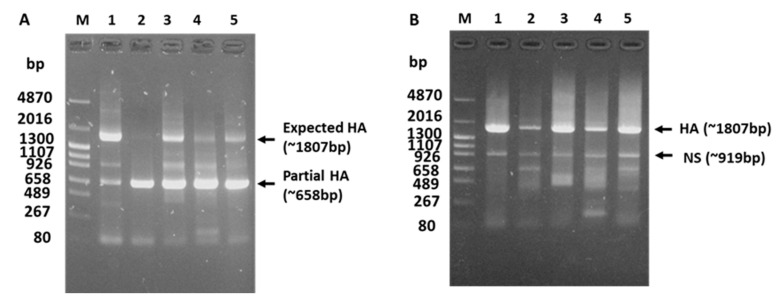
Amplification of the HA genes of H5Nx AIV strains. (**A**) PCR using Hoffmann’s original primers. (**B**) PCR using the modified Hoffmann HA forward primer. 1: A/duck/HongKong/820/80 (H5N3); 2: A/whistling swan/Shimane/499/83 (H5N3); 3: A/chicken/Miyazaki/K11/07 (H5N1); 4: A/Chicken/Yamaguchi/7/04 (H5N1); 5: A/white-tailed eagle/Japan/OU-1/2022 (H5N1); M: DNA size marker.

**Table 1 viruses-15-02274-t001:** Ct values and hemagglutination titers for the white-tailed eagle swab samples.

Swab Sample	Ct Value		Hemagglutination Titer
	Original	AF	AF
Oropharyngeal	36.35	21.81	256
Conjunctival	33.51	20.93	512
Cloacal	34.86	* NT	<2

* NT: Not tested because of the hemagglutination titer was not obtained. The Ct values obtained in real-time RT-PCR of the influenza A virus M gene performed for the original swab samples (Original) and allantoic fluid from the embryonic chicken eggs inoculated with the samples (AF) are presented. The hemagglutination titers obtained for the AF are also presented.

**Table 2 viruses-15-02274-t002:** HI titers of anti-H5 sera and mAbs against homologous and heterologous H5Nx viral strains including WTE/Jp (H5N1).

Virus	Subtype	Year Isolated	Clade	Antiserum			mAbs		
				* Dk/Hk	Ck/Mz	WTE/Hk	3B5.1	3B5.2	1G5
Dk/Hk	H5N3	1980	nonGs/GD	1280	1280	320	1280	160	80
Ck/Ym	H5N1	2004	2.5	640	>5120	80	2560	640	80
Ck/Mz	H5N1	2007	2.2	1280	2560	80	640	640	320
Ws/Hm	H5N1	2011	2.3.2.1	80	160	<40	320	20	<10
WTE/Jp	H5N1	2022	2.3.4.4b	<10	80	1280	<10	<10	<10

* Anti-H5 subtype reference serum provided by the OIE Reference Laboratory. Homologous titers are underlined. Dk/Hk: A/duck/Hong Kong/820/80 (H5N3); Ck/Ym: A/chicken/Yamaguchi/7/04 (H5N1); Ck/Mz: A/chicken/Miyazaki/K11/07 (H5N1); Ws/Hm: A/whooper swan/Hamanaka/2011 (H5N1); WTE/Jp: A/white-tailed eagle/Japan/OU-1/2022 (H5N1); EA-nonGS/GD; Eurasian-origin non-goose/non-Guangdong H5 low pathogenic AIV; WTE/Hk: A/white-tailed eagle/Hokkaido/22-RU-WTE-2/2022.

## Data Availability

The data reported in this paper have been deposited in GenBank, and they are openly available using the accession numbers described in the Results, Section 3.2. Further inquiries can be addressed to the corresponding author.

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
