# Peer review of "Challenges for Precise Subtyping and Sequencing of a H5N1 Clade 2.3.4.4b Highly Pathogenic Avian Influenza Virus Isolated in Japan in the 2022–2023 Season Using Classical Serological and Molecular Methods"

_viruses, 2023, doi:10.3390/v15112274_

Round 1

Reviewer 1 Report (Previous Reviewer 3)

Comments and Suggestions for Authors

Komu et al. presented two challenges and solutions in HA subtyping of the currently circulating HA clade 2.3.4.4b HPAIV by HI testing and HA gene amplification with conventional universal primer. In the re-submitted manuscript, explanations have been added, mainly regarding the limitations due to legal circumstances.

Regarding the antiserum against the reference H5 subtype virus, it is not appropriate to replace the diagnostic antiserum, as it is still effective in detecting common H5 subtype viruses that are not HA clade 2.3.3.4b HPAIVs (Just a comment).

The points I had previously pointed out had been corrected. I think it clearly stated the issues and results that the authors addressed. No further remarks have been made by me. 

Reviewer 2 Report (Previous Reviewer 1)

Comments and Suggestions for Authors

As high costs preclude the use of next-generation sequencing for routine surveillance, classical serological subtyping methods including hemagglutination inhibition and neuraminidase inhibition assays remain reliable, affordable, and accepted serological tools for global influenza surveillance.

Authors corrected shortcomings of the text, made additions and made explanations on questions and comments asked of reviewers.

I will repeat: All influenza researchers are aware of difficulties with routine avian influenza diagnosis. But at present there are few described and proven results of false negative diagnosis. I'm sure this research will get a lot of citations in scientific articles.

This manuscript is a resubmission of an earlier submission. The following is a list of the peer review reports and author responses from that submission.

Round 1

Reviewer 1 Report

Comments and Suggestions for Authors

Because high costs preclude the use of next-generation sequencing for routine surveillance, classical serologic subtyping methods, including hemagglutination inhibition and neuraminidase inhibition assays, remain reliable, affordable, and accepted serologic tools for global influenza surveillance.

All influenza researchers recognize the difficulties of routine diagnosis of avian influenza. However, there are currently no described and proven results of false negative diagnoses. I am sure this study will get many citations in scientific articles.

A very interesting article with a relevant problem. Unfortunately, the web links to the Supplementary Materials do not open. Also, Table 1 and Table 2 are missing in the attached supplementary materials. There are no comments on the rest of the text.

Reviewer 2 Report

Comments and Suggestions for Authors

Dear editor,

thank you for giving me the chance to review Manuscript ID viruses-2619611 entitled Challenges for Precise Subtyping and Sequencing of a H5N1 Clade 2.3.4.4b Highly Pathogenic Avian Influenza Virus Isolated in Japan in the 2022–2023 Season Using Classical Serological and Molecular Methods by Komu et al. for Viruses. The workgroup of Haruko Ogawa has well known experience with in the field of influenza diagnostics and epidemiology and contributed significanty to the field with their previous work. However, their current manuscript describes only fragmentary information of a single case report which is of minor importance to the scientific community. I recommend rejection due to minor importance.

Points of criticism:

1.       The manuscript describes a single case report. However, any information exept the species name are lacking. An acceptable single case report would need at least species name in latin, method of proof of species, signalement including sex, age and weight, anamnestic data including clinical disease, reason of death, and macroscopic and histopathologic postmortal findings.

2.       Why didn’t you analyze individual organs for their virus content? The distribution within the sea-eagle would be highly interesting.

3.       Data concerning the usability of various diagnostic methods should be presented in the context of ring tests or other cross-laboratory and cross-method evaluations with an N of more than one. Especially the relevance for the detected unexpected binding of the pPCR primer needs to be presented in the context of a larger cohort, like all 242 cases of H5-sub-type HPAIV infections in wild birds in Japan between September 25, 2022 and 87 April 19, 2023. Furthermore, a ring test would provide information if this is lab dependent or a general problem.

4.       If this isolate is of special importance, its virulence should be compared to standard isolates using in-vivo tests in chickens by the intravenous pathogenicity index (IVPI) accoriding to the OIE standards.

Reviewer 3 Report

Comments and Suggestions for Authors

To improve your manuscript, please consider the following: 

Introduction

-In line 47, please correct the entry “A/goose/Guangdong/96/Gs/GD” as it does not follow the official strain designation of the influenza virus.

Materials and Methods

-In lines 97-98, I think information on the manufacturer of the viral transport medium is necessary. 

-In lines 118-119, could you please indicate the number of mAb used? The reference no. 21 suggests three mAb, but it should be clarified if there are other mAb antibodies used.

Results

-I wanted to review the contents of the Supplementary Table, but it was not provided for REVIEW. (Just a comment.)

-In lines 158-159, although it is described as Ct values ranging from 33.35 to 36.5, a value of 33.35 is not seen in Table 1. Where did this value come from?

-From the underlining in Table 2, it can be read that the viruses from which 3B5.1, 3B5.2, and 1G5 monoclonal antibodies were generated are Ck/Ym or Ck/Mz. However, it should also be explained in the MM or results text. 

-A/whooper swan/Hamanaka/11 (H5N1) in the Figure 2 description should be written as the original name, A/whooper swan/Hamanaka/2011 (H5N1).

-In line 257, whistling swan may be a mistake for whooper swan (similarly lines 272, 317, 320). 

-In Table 2, can the combination of Anti-WTE/Hk antiserum and WTE/Jp virus be considered homologous? Why is it underlined?

-In Table 2 (line 272), Ws/Hm does not seem to be an abbreviation for A/whistling swan/Shimane/499/83 (H5N3).

Discussion

-I think the authors' arguments are well organized and described. (Just a comment.)

-In lines 324, 325 and 326, STAIN is an error for STRAIN?
